# Text-to-Image Generation Via Energy-Based CLIP

**Roy Ganz**                                                                   *royg27592@gmail.com*
*Electrical Engineering Department*
*Technion*

**Michael Elad**                                                               *elad@cs.technion.ac.il*
*Computer Science Department*
*Technion*

**Reviewed on OpenReview:** *https://openreview.net/forum?id=FBmWiJXIGk*

## Abstract

Joint Energy Models (JEMs), while drawing significant research attention, have not been successfully scaled to real-world, high-resolution datasets. We present CLIP-JEM , a novel approach extending JEMs to the multimodal vision-language domain using CLIP, integrating both generative and discriminative objectives. For the generative one, we introduce an image-text joint-energy function based on Cosine similarity in the CLIP space, training CLIP to assign low energy to real image-caption pairs and high energy otherwise. For the discriminative one, we employ contrastive adversarial loss, extending the adversarial training objective to the multimodal domain. CLIP-JEM not only generates realistic images from text but also achieves competitive results on the compositionality benchmark, outperforming leading methods with fewer parameters. Additionally, we demonstrate the superior guidance capability of CLIP-JEM by enhancing CLIP-based generative frameworks and converting unconditional diffusion models to text-based ones. Lastly, we show that our model can serve as a more robust evaluation metric for text-to-image generative tasks than CLIP.

## 1 Introduction

Energy-based models (EBMs) (LeCun et al., 2006) are a class of models that define a probability distribution over data points using an energy function, where lower energy values correspond to higher probabilities. These models are trained by adjusting the learned function to minimize the energy of observed data points and maximize the energy of synthetic ones, effectively aligning the energy landscape with the true data distribution. Joint Energy Models (JEMs) (Grathwohl et al., 2019) extend EBMs by utilizing a classifier's logits to also model a joint energy function. JEMs are trained with both discriminative and generative objectives, namely, aiming to classify data points and to model the joint energy function, respectively. However, both EBMs and JEMs face significant scalability challenges. Their training processes can be unstable and computationally intensive, restricting their applicability to smaller datasets and making them unsuitable for real-world, high-resolution image datasets.

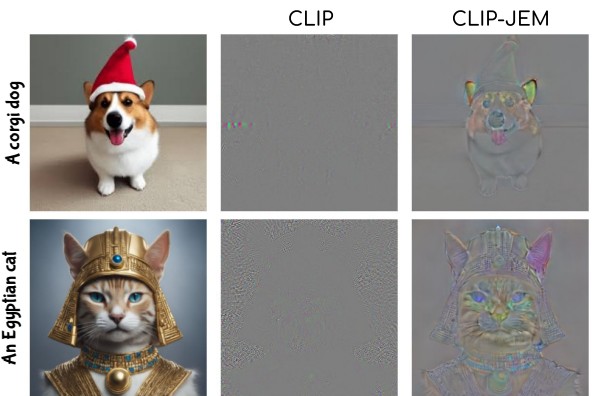

Figure 1: **CLIP-JEM gradients**. Demonstration of the meaningful input gradients of CLIP-JEM compared to a "vanilla" CLIP model with respect to different textual prompts.

Adversarial training (Goodfellow et al., 2015; Madry et al., 2018) is a technique designed to enhance models' robustness against adversarial examples, which are small, imperceptible perturbations added to inputs to mislead classifiers. By training models to correctly classify these adversarial examples, such training results in perceptually aligned gradients (PAG) (Tsipras et al., 2019). PAG refers to the phenomenon where the model's input gradients[1] are semantically meaningful and aligned with human perception, indicating that the features learned by the model are more human-aligned. Recently, the concept of PAG was extended to the multimodal image-text domain with CLIPAG (Ganz & Elad, 2024), which applies adversarial training to the visual part of CLIP (Radford et al., 2021). This approach enables text-based generation through pixel-space optimization. However, while CLIPAG can produce good-looking images, the results are often non-realistic and heavily reliant on multiview augmentation. The need for such augmentation suggests that the gradients themselves are not sufficiently informative to generate realistic images from single views. These limitations highlight the need for more advanced techniques to achieve realistic generation using CLIP models.

In this work, we propose CLIP-JEM, a novel approach that extends JEMs to the multimodal vision-language domain using CLIP by combining it with adversarial training. This combination leverages the strengths of both techniques to address their limitations: mitigating the scalability and stability issues of JEMs and enabling high-resolution text-based generation while overcoming the non-realistic outputs typical of CLIPAG. Inspired by unimodal JEMs, we fine-tune CLIP using two objectives: generative and discriminative. For the generative objective, we introduce an image-text energy function based on Cosine similarity in the CLIP space. We train CLIP to assign low-energy values to real image-text pairs and high values to others. More specifically, inspired by EBMs, we utilize the model to draw text-based generated samples and train it to assign these with high energy values. This is done by an iterative pixel-space optimization following the model's gradients, starting from a random sample. We formulate this as a contrastive loss to align with CLIP. For the discriminative objective, we follow the path set by CLIPAG to define a contrastive adversarial loss. By combining these two objectives, we train the visual encoder of CLIP, resulting in CLIP-JEM, a model with semantically meaningful gradients (fig. 1) capable of generating realistic samples through simple pixel-space optimization (fig. 2).

We establish the effectiveness of CLIP-JEM across three key domains: text-to-image generation, guidance capabilities, and as an evaluation metric. First, CLIP-JEM enables in text-to-image generation through pixel-space optimization, producing realistic images, significantly surpassing CLIPAG by more than 20 FID points, without applying any augmentation. Despite its relatively small size, this model achieves results competitive with much larger models on the challenging compositionality benchmark, CompBench (Huang et al., 2023). Specifically, it surpasses Stable diffusion v2 (Rombach et al., 2022a) and methods tailored for compositionality (Liu et al., 2023b; Feng et al., 2023). We attribute this to the discriminative nature of the model, enabling it to better align with the provided prompts. Furthermore, CLIP-JEM significantly enhances text-based guiding capabilities. To this end, we illustrate that incorporating CLIP-JEM for guidance converts unconditional diffusion models (Dhariwal & Nichol, 2021; Ahn et al., 2024) into text-guided ones with just 25 diffusion steps. Additionally, replacing CLIP with CLIP-JEM in CLIP-based generative frameworks markedly boosts their performance. Finally, CLIP-JEM proves its utility as an evaluation metric (a.k.a. CLIP-Score) for text-based image editing. It shows robustness to adversarial examples and enhanced sensitivity to image quality compared to the "vanilla" counterpart. This indicates that CLIP-JEM is a more reliable and precise tool for assessing the quality and integrity of generated images. To summarize,

- We introduce CLIP-JEM, a novel approach extending Joint Energy Models to the vision-language domain using CLIP.

- CLIP-JEM enables high-resolution text-to-image generation through pixel-space optimization, achieving competitive results on a challenging compositionality benchmark.

- CLIP-JEM enhances text-based guidance capabilities, boosting CLIP-based generative frameworks and converting unconditional diffusion models into text-guided ones.

- We demonstrate that CLIP-JEM can serve as an improved CLIP-Score evaluation metric for text-based image editing.

---

[1]This gradient is computed as the derivative of the chosen output logit w.r.t. the input image.

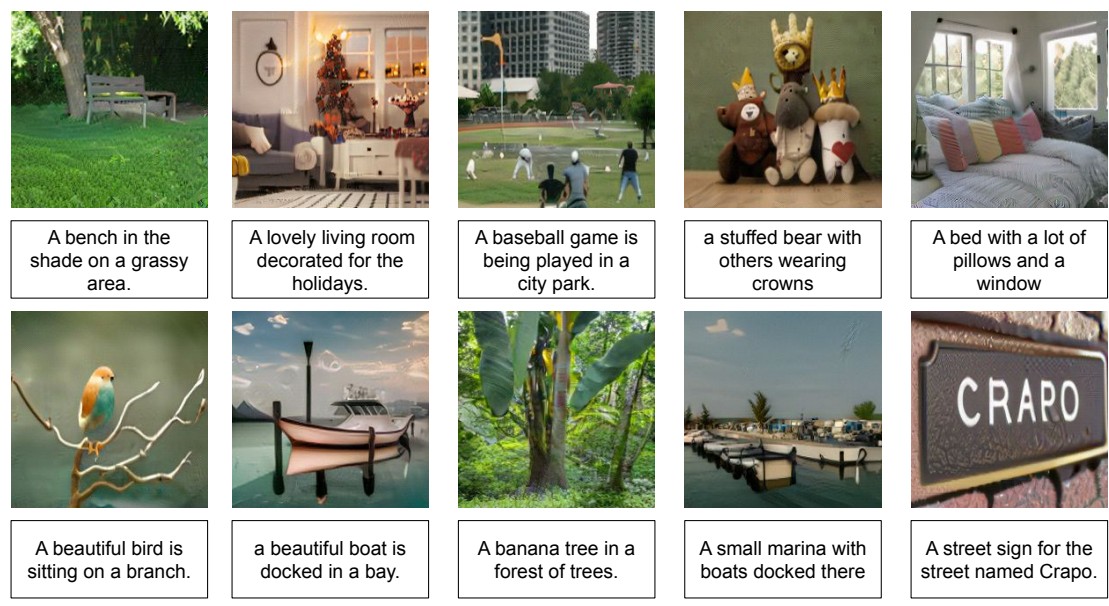

Figure 2: **CLIP-JEM qualitative results**. Images generated using CLIP-JEM with ConvNext-XXL.

## 2 Related Work

### 2.1 Energy-Based Models

EBMs (LeCun et al., 2006) define a probability distribution over data points using an energy function:

$$p_\theta(x) = \frac{\exp(-E_\theta(\mathbf{x}))}{Z(\theta)}, \tag{1}$$

where $E_\theta(\mathbf{x})$ assigns scalar values to data points, and $Z(\theta)$ is a normalizing constant. Lower energy values indicate higher probabilities. Training a model with parameters $\theta$ involves minimizing the energy assigned to "positive" data samples $\mathbf{x}^+ \sim p(\mathbf{x})$ and maximizing the energy of "negative" samples $\mathbf{x}^- \sim p_\theta(\mathbf{x})$. In this framework, the model's ability to "distinguish" between samples is operationalized by assigning lower energy (*i.e.*, higher likelihood) to samples drawn from the data distribution, and higher energy (*i.e.*, lower likelihood) to those drawn from the model distribution. Convergence is achieved when the model can no longer reliably separate positive and negative samples based on their energy values. To sample from $p_\theta(\cdot)$, we employ Stochastic Gradient Langevin Dynamics (SGLD), which begins from a predefined initial distribution and iteratively updates the samples using a step size $\alpha$.

$$\mathbf{x}_{i+1} = \mathbf{x}_i - \frac{\alpha}{2}\frac{\partial E_\theta(\mathbf{x}_i)}{\partial \mathbf{x}_i} + \epsilon, \quad \epsilon \sim \mathcal{N}(\mathbf{0}, \alpha\mathbf{I}). \tag{2}$$

### 2.2 Joint-Energy Models

Recently, Grathwohl et al. (2019) observed that one can parameterized $p_\theta(\mathbf{x}, y)$ and $p_\theta(\mathbf{x})$ using the logits of a classifiers. Given a label $y$ and an image $\mathbf{x}$, the joint distribution can be expressed as

$$p_\theta(\mathbf{x}, y) = \frac{\exp(f_\theta(\mathbf{x})_y)}{Z(\theta)}, \tag{3}$$

where $f_\theta(\mathbf{x})_y$ is the logit corresponding with the $y^{\text{th}}$ class label. Thus, the joint energy function is $E_\theta(\mathbf{x}, y) = -f_\theta(\mathbf{x})_y$. Marginalizing over $y$ results in an unconditional distribution $p_\theta(\cdot)$. Training JEMs involve with optimizing both a discriminative and a generative objectives. Despite having several merits

(*e.g.*, generative capabilities and adversarial robustness), training such models often suffers from instability and even divergence. Despite recent advancements, JEMs perform well for relatively small datasets (mainly SVHN (Netzer et al., 2011), CIFAR (Krizhevsky et al., 2009) and CelebA (Liu et al., 2015)) but are not competitive when brought to real-world visual content (Yang et al., 2023; Zhu et al., 2021; Yang & Ji, 2021). In this work, we aim to extend JEMs into the most challenging setup – text-to-image generation using a CLIP-based model.

### 2.3 CLIP for Text-to-Image Generation

CLIP (Radford et al., 2021) is a vision-language model, pretrained to align a massive corpus of image-text pairs. The outstanding performance of CLIP visual and textual encoders has propelled great advancements in various fields. In Large Vision Language Modeling (Li et al., 2022; Zhu et al., 2023; Liu et al., 2023a; Li et al., 2023; Ganz et al., 2023; 2024), CLIP vision encoder serves as the primary visual backbone, leading to unprecedented performance. In text-to-image generation, two main lines of work harness CLIP: (i) Utilizing CLIP image-text alignment to guide the visual results to be aligned with the textual description (Frans et al., 2022; Crowson et al., 2022; Patashnik et al., 2021; Gal et al., 2022; Kwon & Ye, 2022; Vinker et al., 2022); and (ii) Using CLIP's text encoder to condition generative models (Kang et al., 2023a; Nichol et al., 2022; Ramesh et al., 2022; Rombach et al., 2022b). Unlike these works, which utilize CLIP along with a generative model, we aim to cast CLIP into an energy-based model, capable of performing text-to-image generation without an additional generative model.

### 2.4 Perceptually Aligned Gradients

Adversarially robust models (Carlini & Wagner, 2017; Madry et al., 2018) are designed to withstand adversarial attacks (Szegedy et al., 2014; Goodfellow et al., 2015), which are small, imperceptible perturbations aimed at misleading classifiers. It has been observed that such models exhibit a phenomenon known as Perceptually Aligned Gradients (PAG), which is absent in their non-robust counterparts. PAG refers to the model's input gradients being semantically meaningful and aligned with human perception, indicating that the features learned are more aligned with human vision (Ilyas et al., 2019; Engstrom et al., 2019; Salman et al., 2020). PAG has been harnessed for generative tasks, such as image generation and image-to-image translation (Santurkar et al., 2019), thereby improving state-of-the-art image synthesis results (Ganz & Elad, 2022). PAG has also been explored for enhanced robust classification (Blau et al., 2023). Recently, the study of PAG has been extended to the multimodal domain using CLIPAG (Ganz & Elad, 2024), which applies adversarial training to the multimodal text-to-image domain using CLIP (Radford et al., 2021). This approach enables text-based generation through pixel-space optimization. However, while CLIPAG can produce good-looking images, the results are often non-realistic and heavily reliant on multiview augmentation. The need for such augmentation suggests that the gradients themselves are not sufficiently informative to generate realistic images from single views. These limitations highlight the need for more advanced techniques for realistic text-to-image generation using CLIP models.

## 3 Method

In this work, we aim to extend Joint Energy Models to the challenging text-to-image generation setting using a CLIP model. Our overall framework, illustrated in Figure 3, consists of two main objectives: a contrastive energy loss and an adversarial loss. In section 3.1, we first define the `joint image-text energy` function, forming a measure of the faithfulness and alignment of the given pair, and elaborate on its training procedure. Next, in section 3.2, we detail the contrastive adversarial loss, extending CLIP's loss to the adversarial case. Lastly, in section 3.3, we describe the overall training procedure of CLIP-JEM, resulting in a multimodal energy model capable of text-to-image generation.

### 3.1 Joint Image-Text Energy Via CLIP

We extend a pretrained CLIP to model the joint energy of image-text pairs. We denote its vision and textual towers as $f_\theta^I$ and $f_\theta^T$, respectively. Given these notations, we propose to utilize CLIP to formulate an

image-text joint distribution,

$$p_\theta(\mathbf{I}, \mathbf{T}) = \frac{\exp(\mathrm{CosineSimilarity}(f_\theta^I(\mathbf{I}), f_\theta^T(\mathbf{T})))}{Z(\theta)}, \tag{4}$$

where $\mathbf{I}$ and $\mathbf{T}$ are visual and textual inputs, `CosineSimilarity` is the well-known Cosine similarity measure used by CLIP, and $Z(\theta)$ is an unknown normalizing factor. Thus, the induced joint image-text energy function is $E_\theta(\mathbf{I}, \mathbf{T}) = -\mathrm{CosineSimilarity}(f_\theta^I(\mathbf{I}), f_\theta^T(\mathbf{T}))$, where a higher degree of image-text similarity results in lower energy values and higher probability and vice-versa.

Similar to energy-based model training, adapting CLIP to form an energy measure requires both "positive" and "negative" image-text pairs, denoted as $(\mathbf{I}, \mathbf{T})$ and $(\tilde{\mathbf{I}}, \mathbf{T})$, respectively. As CLIP is a contrastively-trained model, we utilize such pairs to formulate a contrastive energy objective, illustrated on the upper-left side of Figure 3. This rectangular matrix contains the negative joint-energy values of textual and visual inputs. Specifically, the upper part contains the energy of the "positive" image inputs and the lower one the "negatives". Given this contrastive energy matrix, we train the model weights of $f_\theta^I(\cdot)$ using the cross-entropy loss with the main diagonal, marked in blue, as the ground-truth annotations. Namely, the objective is to obtain high values in this main diagonal (low energy, high probability) and low values elsewhere. Focusing on a certain textual input $\mathbf{T}_k$, minimizing the proposed contrastive energy loss results in high Cosine similarity values of the "positive" pair $(\mathbf{I}_k, \mathbf{T}_k)$ and low ones for the "negative" pair $(\tilde{\mathbf{I}}_k, \mathbf{T}_k)$. This formulated loss has two additional advantages: First, minimizing it results in providing low Cosine similarities and hence high energy and low probability for unmatching image-text pairs $(\mathbf{I}_i, \mathbf{T}_j)$ for $i \neq j$, enhancing the model's discriminative ability. Second, it enables the utiliza-

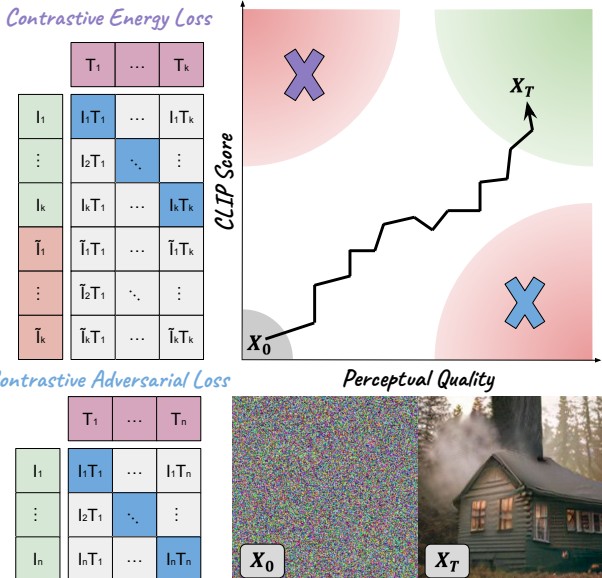

Figure 3: **CLIP-JEM method**. Illustration of the contrastive losses in our method and their effect on enabling realistic text-based image synthesis via pixel-space optimization.

tion of different realizations of negative samples per each text prompt, which we empirically find to stabilize the training.

The remaining question is how to obtain the "negative" samples. Similar to the methodology in the JEM line of work, we craft such samples in an iterative process based on the model's gradients, starting from a simple canonical distribution (*e.g.*, a Unifrom distribution). Current JEM works utilize SGLD (Welling & Teh, 2011), which requires hundreds of iterations to obtain good samples. To mitigate the computation overhead involved in generating such samples, such works utilize a replay buffer. This data structure stores thousands of generated "negative" samples and updates them throughout the training process, enabling a relatively small number of SGLD steps. However, in the image-text context, employing such a solution is not feasible, as, unlike multiclass datasets such as CIFAR-10, which have a fixed amount of classes, the text-to-image setting is open-vocabulary in nature with practically infinite number of possible captions. Thus, in our setting, maintaining a replay buffer and updating its samples is not a practical solution, as the probability of having the exact two captions in a dataset is very low, eliminating the usefulness of such a structure. To enable fast "negative" sampling, we propose to avoid employing SGLD and utilize a momentum-based optimization with an adaptive learning rate instead. In practice, we use AdamW optimizer (Loshchilov & Hutter, 2017) to update the image in the iterative process, which enables drawing samples within a relatively small number of steps. Overall, we first draw the initial sample from a canonical uniform distribution, $\tilde{I}^{t=0}$. Next, we compute the Cosine similarity between the visual and textual encodings and calculate its input-gradients. Lastly, we update the visual inputs accordingly and repeat the process for $T_{\mathrm{JEM}}$ iterations. Notably, we calculate the

gradients on a slightly noisy version of the visual input, introducing randomness to the sampling mechanism. In the supplementary materials, we discuss the connection of our sampling procedure to SGLD.

## 3.2 Contrastive Adversarial Training

Similar to CLIPAG (Ganz & Elad, 2024), we extend adversarial training to the contrastive loss, forming a contrastive adversarial loss. Given an input batch of image-text pairs, we perform a two-step procedure – (i) crafting visual adversarial examples that maximizes the similarity w.r.t the matching texts; (ii) optimizing the vision encoder's weight to minimize the contrastive adversarial loss. The adversarial contrastive loss matrix is illustrated on the lower-left side of Figure 3, and in eq. (5). In this matrix, the vertical green vector represents the encodings of adversarial visual input, while the horizontal pink vector represents the textual encodings. This approach ensures that the model learns to be robust against adversarial perturbations and equips it with semantically meaningful gradients (*i.e.*, PAG), which enhance the iterative generation process.

$$\min_{\theta_I, \theta_T} \sum_{(\mathbf{x}, t) \in \mathcal{D}} \max_{\delta \in \Delta} (1 - \text{CosineSimilarity}(f_{\theta_I}^I(\mathbf{x} + \delta), f_{\theta_T}^T(t))). \tag{5}$$

## 3.3 Training Procedure

CLIP-JEM's objective is the combination of contrastive adversarial and contrastive energy losses, and the overall training protocol is described in the supplementary materials (algorithm 1). The goal of CLIP-JEM is to obtain an image-text energy model capable of generating images based on textual descriptions. On the right side of Figure 3, we illustrate this generation process (piece-wise linear arrow), starting from a random sample from a canonical distribution (marked in gray). In this figure, any image-text pair is represented by two values: CLIP score and perceptual quality. The former measures the alignment of the pair according to our CLIP model, while the latter is a conceptual measure of the image's visual quality. We aim to generate samples with high CLIP scores (low energy) and good perceptual quality. We analyze the contribution of both objectives to this goal. The contrastive energy loss prevents CLIP-JEM from assigning high CLIP scores to low-quality images, as throughout the energy training, the model is trained to lower the CLIP scores of "negative" samples and provide high scores to the "positive" ones, which have high perceptual quality. The contrastive adversarial objective trains the model against adversarial inputs, namely, imperceptible visual changes that significantly decrease the CLIP score (these samples reside in the bottom right corner of the figure's center part). Thus, this loss prevents CLIP-JEM from assigning low CLIP scores to high-quality inputs. Overall, combining the two objectives eliminates the red modes in the upper-right panel of Figure 3, preventing the iterative generation from drawing such samples. This, in return, leads to drawing samples of the green mode, which are realistically looking images that align with their corresponding text. We demonstrate the contribution of combining these two objectives in the appendix D.

## 4 Experiments

We train different variants of CLIP using Algorithm 1 – including ViT-B/32 and ConvNext in base, large, and XXL configurations on the extensive image-caption DataComp dataset (Gadre et al., 2024) for $20,000$ steps. Throughout the training process, we keep the text encoder frozen and update solely the vision encoder. Implementation and training details are provided in the supplementary materials. To analyze the performance of CLIP-JEM, we first evaluate it in the text-to-image generation setting Section 4.1). Next, we demonstrate its effectiveness as a guiding model (Section 4.2). Lastly, we show that CLIP-JEM can serve as an improved evaluation metric compared to the "vanilla" CLIP, attributed to its robustness and awareness of perceptual quality.

## 4.1 Text-To-Image Generation

Similar to the training procedure, we perform pixel-space optimization to generate samples. Given a target prompt, we initialize the image as a random sample from a uniform distribution and perform 50 steps to

Table 1: **MS-COCO text-to-image generation results**. Frechet Inception Distance (FID, lower is better) and CLIPSIM (higher is better) results, along with model sizes. "ZS" indicates whether the model was trained on MS-COCO.

| Method | #Params. | ZS | FID↓ | CLIPSIM↑ |
|---|---|---|---|---|
| Stack-GAN | - | ✗ | 74.1 | - |
| AttnGAN | 230M | ✗ | 35.5 | 27.7 |
| CogView | 4,000M | ✗ | 27.1 | 33.2 |
| DALL-E | 12,000M | ✓ | 27.5 | - |
| GLIDE | 6,000M | ✓ | 12.2 | - |
| LDM-KL-8 | 1,450M | ✓ | 23.3 | - |
| LDM-KL-8-G | 1,450M | ✓ | 12.6 | - |
| LAFITE | 226M | ✓ | 26.9 | - |
| StyleGAN-T | ~1100M | ✓ | 13.9 | - |
| NÜWA | 870M | ✓ | 12.9 | 34.3 |
| GigaGAN | 1034M | ✓ | **9.1** | - |
| CLIPAG$_L^\dagger$ | 200M | ✓ | 82.0 | 30.3 |
| CLIPAG$_L$ | 200M | ✓ | 47.6 | 33.4 |
| CLIP-JEM$_{ViT}$ | 88M | ✓ | 68.3 | **34.5** |
| CLIP-JEM$_B$ | 88M | ✓ | 34.8 | 31.6 |
| CLIP-JEM$_L$ | 200M | ✓ | 26.7 | 31.7 |
| CLIP-JEM$_{XXL}$ | 846M | ✓ | 23.4 | 33.5 |

maximize the cosine similarity with respect to the text, using an AdamW optimizer with no momentum. We evaluate the performance of CLIP-JEM in two main setups: image quality and compositionality.

**Quality and Fidelity** We use CLIP-JEM to generate $30,000$ samples from the MS-COCO dataset (Lin et al., 2015) and report the results in FID[2] and CLIPSIM using ViT-B/32 in Table 1. We compare CLIP-JEM to various GAN-based, diffusion-based, and autoregressive text-to-image models (see more details in the supplementary materials). We report the number of parameters of the baselines and the size of the vision encoder used for CLIP-JEM . As shown in Table 1, scaling up the model size significantly benefits our method, substantially improving the FID scores. Specifically, in the XXL case, CLIP-JEM performs similarly to the unguided Latent Diffusion Model (LDM-KL-8) and outperforms DALL-E and CogView despite being smaller. Additionally, we train a CLIPAG baseline using the same training configuration and architecture to better demonstrate the effectiveness of our image-text energy objective. We report the results of two variants of CLIPAG using ConvNext Large – with and without multiview augmentations, denoted as CLIPAGL and CLIPAGL$^\dagger$, respectively. As can be seen in CLIPAG results, the multiview augmentation pipeline is crucial (improves the FID from 82.0 to 47.6), highlighting the unsatisfying quality of its gradients. Interestingly, using the same model with CLIP-JEM leads to a much-improved FID score (26.7) without applying any augmentation. This strongly indicates the effectiveness of introducing our contrastive energy loss and our method's improved capability of generating realistic samples. In the CLIPSIM metric, CLIPAG$_L$ leads to a better result than CLIP-JEM . We attribute this to the fact that the multiview augmentations lead to unrealistic images (which impair the FID) that highly align with the text (increasing the CLIPSIM). Overall, these results indicate that CLIP-JEM achieves both of its goals – extending JEM training into text-to-image generation and improving CLIPAG's photorealism.

**Compositionallity** We compare CLIP-JEM with other generative models using T2I CompBench (Huang et al., 2023), which evaluates open-world compositional text-to-image generation across attribute binding (color, shape and texture), object relationship (spatial and non-spatial) and complex. Despite advances in text-to-image generation, models still struggle to compose objects with different characteristics and

---

[2]We use the same evaluation codes with DM-GAN, which is available at `https://github.com/MinfengZhu/DM-GAN`

Table 2: **T2I-CompBench results**. CLIP-JEM performance on the text-to-image generation compositionality benchmark.

| | Model | Attribute Binding | | | Object Relationship | | Complex | Average |
|---|---|---|---|---|---|---|---|---|
| | | Color ↑ | Shape ↑ | Texture ↑ | Spatial ↑ | Non-spatial ↑ | | |
| | SD1.4 | 0.3765 | 0.3576 | 0.4156 | 0.1246 | 0.3079 | 0.3080 | 0.3150 |
| | SD2 | 0.5065 | 0.4221 | 0.4922 | 0.1342 | 0.3127 | 0.3386 | 0.3677 |
| | Composable (SD2) | 0.4063 | 0.3299 | 0.3644 | 0.0800 | 0.2980 | 0.2898 | 0.2947 |
| | Structured (SD2) | 0.4990 | 0.4218 | 0.4900 | 0.1386 | 0.3111 | 0.3355 | 0.3660 |
| | Attn-Exct (SD2) | **0.6400** | 0.4517 | 0.5963 | **0.1455** | 0.3109 | **0.3401** | **0.4141** |
| CLIP-JEM | CLIP-JEM$_{ViT}$ | 0.5305 | 0.5159 | 0.5566 | 0.0262 | **0.3343** | 0.2900 | 0.3756 |
| | CLIP-JEM$_B$ | 0.5799 | 0.5122 | **0.6154** | 0.0708 | 0.3145 | 0.2938 | 0.3978 |
| | CLIP-JEM$_L$ | 0.5715 | **0.5202** | 0.6072 | 0.0768 | 0.3152 | 0.3020 | 0.3988 |
| | CLIP-JEM$_{XXL}$ | 0.5670 | 0.5021 | 0.6132 | 0.0841 | 0.3205 | 0.3129 | 0.4000 |

relationships into a coherent image. Following CompBench procedure, we generate 10 samples per prompt and average the results on the validation sets of each category, using the same evaluation metrics as in the original paper (B-VQA, UniDet, CLIP, and 3-in-1 for the attribute binding, spatial, non-spatial, and complex categories). Table 2 reports CLIP-JEM 's results compared to top-performing models, including an average score across six categories. CLIP-JEM excels in attribute binding, associating attributes with corresponding objects in generated images, and performs well in the non-spatial relationship category (*e.g.*, "speak to" and "look at"). However, it scores lower in the spatial relationship category due to CLIP's known limitations in spatial compositionality (Yuksekgonul et al., 2022; Lewis et al., 2022). In the complex category, involving multiple objects and attributes, CLIP-JEM performs well despite containing spatial relationships, due to its strengths in attribute binding and non-spatial understanding. Overall, CLIP-JEM outperforms most baselines in compositional generation, despite being smaller. Specifically, it surpasses methods deliberately designed to tackle compositionality and rely on a much stronger generative model (Stable Diffusion v2). We attribute this success to the generative and discriminative objectives combination, enabling effective alignment with compositional prompts.

## 4.2 Text Guidance Using CLIP-JEM

As shown in fig. 1, CLIP-JEM possesses semantically meaningful gradients with respect to a given text. In this section, we demonstrate the guidance capability of our method in two main settings: diffusion guidance and improving CLIP-based generative frameworks, utilizing a ConvNext Large model.

**Diffusion guidance** We utilize CLIP-JEM as a guiding technique to transform unconditional diffusion models trained on ImageNet (Dhariwal & Nichol, 2021; Ahn et al., 2024) into text-conditioned ones using only 25 DDIM steps 25 (Song et al., 2020). In each DDIM step $t$, we update the estimations for the clean image $\hat{\mathbf{x}}_0$ using the gradients of CLIP-JEM (eq. (6)) and use it to compute $\mathbf{x}_{t-1}$ and continue the reverse DDIM process.

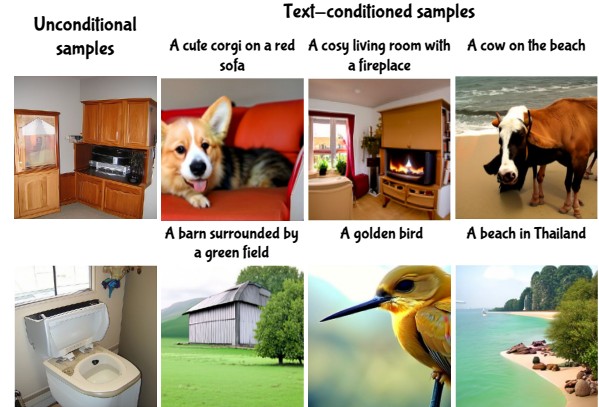

Figure 4: **Diffusion guidance using CLIP-JEM**. Converting an unconditional diffusion model into a text-based one with CLIP-JEM. In each row, we plot the unconditional alongside the guidance results using the same seed.

$$\hat{\mathbf{x}}_0 = \hat{\mathbf{x}}_0 + s \cdot \nabla_{\hat{\mathbf{x}}_0} \text{CosineSimilarity}(f_\theta^I(\hat{\mathbf{x}}_0), f_\theta^T(\mathbf{T})) \qquad (6)$$

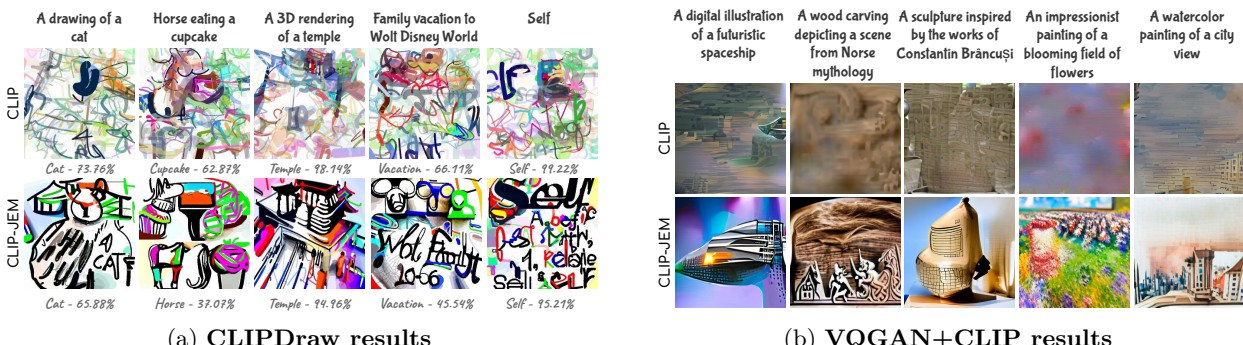

(a) **CLIPDraw results**    (b) **VQGAN+CLIP results**

Figure 5: **Improving CLIP-Based Generative Frameworks via CLIP-JEM**. Qualitative results of CLIP-JEM compared to CLIP using CLIPDraw and VQAGAN+CLIP in a zero-shot setting.

We provide qualitative results of text-based diffusion guidance in Figure 4, showcasing the capability of CLIP-JEM to convert unconditional diffusion models to text-based ones, thereby enhancing their utility.

**Improving CLIP-Based Generative Frameworks**    CLIP is widely used in text-to-image generation frameworks to update the resulting image to better align with the target text in CLIP's space. This is typically done using the input gradients of the CLIP's vision encoder with respect to the text (maximizing the cosine similarity). However, these gradients tend to be semantically meaningless, as demonstrated in Figure 1. To mitigate this, a multiview augmentation pipeline is often employed to acquire semantically meaningful gradients. Our method, on the other hand, inherently produces gradients that convey rich semantic information. Consequently, using CLIP-JEM can eliminate the need for multiview augmentations, thereby improving computational efficiency. To demonstrate this, we experiment with two such frameworks: CLIPDraw (Frans et al., 2022) and VQGAN+CLIP (Crowson et al., 2022). CLIPDraw generates drawings by optimizing the parameters of Bézier curves using CLIP, and VQGAN+CLIP updates the latent code of a VQGAN to enable text-to-image generation. In both cases, we do not perform multiview augmentations and report the results with augmentations in the supplementary materials. In Figure 5a, we present the results of CLIPDraw on target texts from the original paper, along with CLIP's top prediction for each prompt. As shown, using CLIP-JEM leads to significantly better visual results, which are more aligned with the text than those generated by the "vanilla" CLIP. The high percentages in the top predictions imply that the images generated by CLIP have an adversarial nature, maximizing the score without performing significant modifications. Next, we evaluate the effect of using our approach to guide VQGAN. To this end, we prompted ChatGPT to provide artistic target texts, aligning with the original paper's domain. As seen in Figure 5b, removing the augmentation pipeline results in non-meaningful outputs, whereas CLIP-JEM generates semantically meaningful images. These results strongly attest to the improved guidance capabilities of CLIP-JEM.

## 4.3    CLIP-JEM as an Evaluation Metric

CLIP is often used to evaluate text-to-image generative tasks by measuring cosine similarity between textual descriptions and images in its embedding space, known as CLIP-T. We compare CLIP-JEM with the standard CLIP model, focusing on the ViT-B/32 commonly used for this purpose. Using TEdBench (Kawar et al., 2023), we evaluate CLIP-T scores for various inputs, including outputs from a top-performing generative model (Imagic), source images ("No Edit"), and "Noise" images. Results are shown in table 3 under "Vanilla". We also assess robustness under adversarial attacks with a low perturbation budget ($\epsilon = 2/255$). These attacks aim to increase scores for "bad" images and decrease scores for "good" ones. Our findings indicate that CLIP is highly susceptible to adversarial attacks, resulting in higher CLIP-T scores for non-edited and noise images than for Imagic outputs. In contrast, CLIP-JEM remains robust, maintaining higher scores for Imagic outputs even under adversarial attacks.

Table 3: **Robustness To Adversarial Perturbation**.

| Input images | CLIP | | CLIP-JEM | |
| --- | --- | --- | --- | --- |
| | Vanilla | Attack | Vanilla | Attack |
| Imagic | **0.3031** | 0.2053 | **0.2016** | **0.1951** |
| No Edit | 0.2740 | **0.3547** | 0.1811 | 0.1866 |
| Noise | 0.2033 | 0.3037 | 0.0905 | 0.0959 |

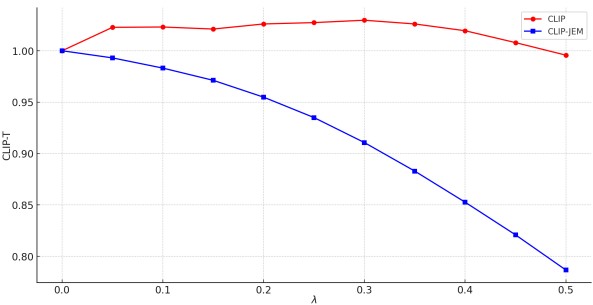

Figure 6: **Sensitivity to perceptual quality**.

We further analyze the effect of perceptual quality on CLIP-T scores by blending Imagic images ($\mathbf{x}_{Imagic}$) with uniform noise: $\lambda\mathbf{x}_{Imagic} + (1 - \lambda)\mathbf{u}$ where $\mathbf{u} \sim \mathcal{U}[0, 1]$. We plot CLIP-T scores for varying $\lambda$ values in Figure 6, normalizing scores to 1.0 at $\lambda = 0$. While the standard CLIP model prefers noisier versions up to $\lambda = 0.45$, CLIP-JEM 's scores decrease with increasing noise, indicating greater sensitivity to image quality, attributed to the contrastive energy objective making the model to assign high energy to non-real images.

## 5 Discussion and Conclusion

In this work, we introduce CLIP-JEM, a novel approach that extends Joint Energy Models to the multimodal domain using CLIP. Through extensive evaluations, CLIP-JEM demonstrates its ability to generate high-quality, compositionally coherent images, achieving competitive results on the MS-COCO dataset and excelling in the T2I CompBench benchmark. Moreover, CLIP-JEM showcases strong guiding capabilities, significantly improving the performance of CLIP-based generative frameworks and converting unconditional diffusion models to text-based ones. Additionally, CLIP-JEM proves to be a robust and perceptually aware evaluation metric, maintaining high scores under adversarial attacks and showing greater sensitivity to image quality than the standard CLIP model. We hope that the insights and findings presented in this paper will inspire further exploration and advancements in multimodal JEM research.

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

# A Detailed background on energy-based models

**Energy based models** Energy-based models (EBMs) (LeCun et al., 2006) are a class of probabilistic models that define a probability distribution over the data points via an energy function. Specifically, EBMs utilize the fact that any probability density $p(\mathbf{x})$ for $\mathbf{x} \in \mathbb{R}^D$ can be written as

$$p_\theta(x) = \frac{\exp(-E_\theta(\mathbf{x}))}{Z(\theta)}, \tag{7}$$

where $E_\theta(\cdot)$ is the *energy function*, which assigns scalar values to data points ($E_\theta : \mathbb{R}^D \rightarrow \mathbb{R}$) and $Z(\theta)$ is a normalizing constant. In EBMs, the probability of a given data point is determined by its energy, where lower energy indicates a higher probability, as can be seen in Equation (7).

Training such a model involves learning the parameters $\theta$ of the energy function, typically modeled by a neural network, to minimize the energy of observed "positive" data points, $\mathbf{x}^+ \sim p(\mathbf{x})$, while maximizing the energy of "negative" samples, $\mathbf{x}^- \sim p_\theta(\mathbf{x})$, generated by the model during training. The training converges when the model cannot detect whether a sample is "positive" or "negative", attesting that $p_\theta$ and $p$ have a similar distribution.

EBMs require sampling from $p_\theta$, both during training to acquire "negative" samples and during inference to synthesize new images. In practice, drawing such samples is done via Stochastic Gradient Langevin Dynamics (SGLD), initialized with $\mathbf{x}_0 \sim p_0(\mathbf{x})$ from a canonical distribution (e.g. Uniform over the input domain), and updated iteratively by

$$\mathbf{x}_{i+1} = \mathbf{x}_i - \frac{\alpha}{2} \frac{\partial E_\theta(\mathbf{x}_i)}{\partial \mathbf{x}_i} + \epsilon, \quad \epsilon \sim \mathcal{N}(\mathbf{0}, \alpha\mathbf{I}), \tag{8}$$

where $\alpha$ is the sampler's step size.

**Joint energy models** Recently, Grathwohl et al. (2019) observed that one can utilize a classifier to model an *energy function* and train it accordingly. Given a classifier $f_\theta$ which maps inputs into K values, known as logits ($f_\theta : \mathbb{R}^D \rightarrow \mathbb{R}^K$), one can parameterize a conditional distribution using the `Softmax` function:

$$p_\theta(y|\mathbf{x}) = \frac{\exp(f_\theta(\mathbf{x})_y)}{\sum_{y'} \exp(f_\theta(\mathbf{x})_{y'})}, \tag{9}$$

where $f_\theta(\mathbf{x})_y$ is the logit corresponding with the $y^{\text{th}}$ class label. The key observation is that one can construct expressions for $p_\theta(\mathbf{x}, y)$ and $p_\theta(\mathbf{x})$ using the classifier's logits. The joint distribution of a data point $\mathbf{x}$ and a label $y$ is given via

$$p_\theta(\mathbf{x}, y) = \frac{\exp(f_\theta(\mathbf{x})_y)}{Z(\theta)}, \tag{10}$$

where $Z(\theta) = \int_\mathbf{x} \exp(f_\theta(\mathbf{x})_y) d\mathbf{x}$ is an intractable normalizing factor. Accordingly, we can define a joint energy function $E_\theta(\mathbf{x}, y) = -f_\theta(\mathbf{x})_y$. Using marginalization over $y$, we can obtain the unconditional distribution,

$$p_\theta(\mathbf{x}) = \sum_y p_\theta(\mathbf{x}, y) = \frac{\sum_y \exp(f_\theta(\mathbf{x})_y)}{Z(\theta)}. \tag{11}$$

Thus, in the unconditional case, the energy function is given as $E_\theta(\mathbf{x}) = -\log \sum_y \exp(f_\theta(\mathbf{x})_y)$.

With this interpretation, one can train a joint model for both discriminative and generative modeling, optimizing both classification and EBM objectives. Such models lead to good classification capabilities, showcasing impressive adversarial robustness while being able to generate new data samples. Nevertheless, training such models often suffers from instability and even divergence, making it applicable mainly for small datasets but not for real-world ones.

| Arch. | BS disc. | BS gen. | #Steps | LR | WD | Sched. | Warmup | Adv. $\epsilon$ | $T_{\text{adv}}$ | $T_{\text{JEM}}$ | $\gamma$ | $\alpha_1$ | $\alpha_2$ |
|---|---|---|---|---|---|---|---|---|---|---|---|---|---|
| ViT-B/32 | 256 | 32 | | $2 \times 10^{-5}$ | | | | | | | | | |
| ConvNext-B | 128 | 32 | | $2 \times 10^{-5}$ | | | | | | | | | |
| ConvNext-L | 128 | 16 | 20K | $2 \times 10^{-5}$ | $1 \times 10^{-4}$ | Cosine | 200 | 3.0 | 5 | 50 | 0.1 | 1.5 | 0.025 |
| ConvNext-XXL | 32 | 8 | | $2 \times 10^{-6}$ | | | | | | | | | |

Table 4: **Implementation details**. We provide the training hyperparameters of CLIP-JEM for the different architectures (BS disc. and gen. stands for the discriminative and generative batch sizes, respectively).

| Arch. | Time [Sec.] | Memory [M] |
|---|---|---|
| ViT-B/32 | 1.4 | 2787 |
| ConvNext-B | 2.6 | 3009 |
| ConvNext-L | 2.4 | 4523 |
| ConvNext-XXL | 4.5 | 11837 |

Table 5: **Sampling time and memory**. The time per-sample in seconds and memory consumption of the different considered models.

## B    Implementation details

**Training hyperparameters**    We implement our method upon the OpenClip codebase[3]. We consider the following model architectures from the model zoo – convnext_xxlarge, convnext_large_d, convnext_base_w and ViT-B-32 with the following pretrained weights, respectively – laion2b_s34b_b82k_augreg_soup, laion2b_s26b_b102k_augreg, laion2b_s13b_b82k_augreg and openai. In table 4, we report the training hyperparameters for CLIP-JEM . We use these hyperparameters to train the different architectures on DataComp. To train our models, we use 8 A40 GPUs for training. Training the largest variant (ConvNext-XXL) for 20K iterations takes 10 days. During training, in the generation process of the "negative" samples we employ a momentum of 0.9 to the AdamW optimizer. However, throughout the inference phase, we do not utilize momentum at all. We aim to make our code and pretrained models publicly available upon acceptance.

**Experimental settings**    To measure the quality and fidelity of the generated images of our method, we compare it to strong baselines using MS-COCO dataset (table 1). Specifically, we compare CLIP-JEM to text-to-image generative models of different types: (i) GAN-based – Stack-GAN (Zhang et al., 2017), AttnGAN (Xu et al., 2017), LAFITE (Zhou et al., 2022), StyleGAN-T (Sauer et al., 2023), and GigaGAN (Kang et al., 2023b) (ii) Diffusion-based – GLIDE (Nichol et al., 2022), and LDM (Rombach et al., 2022c), and Autoregressive ones – DALL-E (Ramesh et al., 2021), CogView (Ding et al., 2021), and NÜWA (Wu et al., 2021). Our reported results and model sizes originate from the respective papers. The CLIPAG baseline results were obtained by us, by removing the contrastive energy loss term, using the same architecture and hyperparameters. As for the CompBench results (table 2), we report the one from the benchmark's paper.

As for the improving CLIP-based generative frameworks experiments, we use an ADM-based model (Dhariwal & Nichol, 2021) with perturbed-attention guidance (Ahn et al., 2024) which strongly improves the unconditional generation as our baseline for diffusion guidance.

**Sampling time and memory**    We train 4 different architectures of CLIP-JEM. As different model size and structure affects the runtime complexity, we report the time of our generation process using a batch size of 1 using an Nvidia A40 GPU in table 5. As expected, increasing the model size leads to more memory consumption and increases the time per sample. However, the ConvNext-B generation time is slightly larger than the ConvNext-L. This is due to the fact the we utilize the wide variant of the ConvNext-B, which is not available in ConvNext-L.

---

[3]https://github.com/mlfoundations/open_clip

---

**Algorithm 1 CLIP-JEM Training**. Given CLIP image and text encoders $f_\theta^I(\cdot)$ and $f_\theta^T(\cdot)$, image-text dataset $\mathcal{D}$, adversarial budget $\epsilon$, adversarial and energy step-sizes $\alpha_1, \alpha_2$, energy loss coefficient $\gamma$, and number of adversarial and generation iterations $T_{\text{adv}}, T_{\text{JEM}}$:

---

**while** *not converged* **do**

    Sample $(\mathbf{I}, \mathbf{T})$ from dataset $\mathcal{D}$

    /* Contrastive adversarial loss                                                  */

    $\delta_0 \leftarrow \mathbf{0}$ **for** *t from 0 to $T_{adv}$* **do**

        $\delta_{t+1} = \Pi_\epsilon(\delta_t + \alpha_1 * \text{ClipLoss}(f_\theta^I(\mathbf{I} + \delta_t), f_\theta^T(\mathbf{T})))$

    **end**

    $\mathbf{I}_{\text{adv}} = \mathbf{I} + \delta_{T_{\text{adv}}}$

    $\mathcal{L}_{\text{adv}} = \text{ClipLoss}(f_\theta^I(\mathbf{I}_{adv}), f_\theta^T(\mathbf{T}))$

    /* Contrastive energy loss                                                     */

    Sample initial "negative" sample $\tilde{\mathbf{I}}$.

    Optimizer $\leftarrow$ AdamW(params $= \tilde{\mathbf{I}}, \text{lr} = \alpha_2$)

    **for** *t from 0 to $T_{JEM}$* **do**

        $\mathcal{L}_{\text{JEM}} = \text{ClipLoss}(f_\theta^I(\tilde{\mathbf{I}} + \beta\mathbf{n}), f_\theta^T(\mathbf{T}))$

        /* $\mathbf{n} \sim \mathcal{N}(0, \mathbf{I})$,    $\beta$ is a small scalar                               */

        Calculate $\partial\mathcal{L}_{\text{JEM}}/\partial\tilde{\mathbf{I}}$ and perform an optimizer step

    **end**

    $\mathcal{L}_{\text{JEM}} = \text{ClipLoss}(f_\theta^I(\text{Concat}(\mathbf{I}, \tilde{\mathbf{I}})), f_\theta^T(\mathbf{T}))$

    /* Update the vision encoder                                               */

    $\mathcal{L} = \mathcal{L}_{\text{adv}} + \gamma * \mathcal{L}_{\text{JEM}}$

    Calculate $\partial\mathcal{L}/\partial\theta$ and update CLIP image encoder $f_\theta^I(\cdot)$

**end**

---

## C   Training protocol

In algorithm 1, we detail to overall training procedure of CLIP-JEM. First, we draw image-text batches from dataset $\mathcal{D}$ and apply $T_{\text{adv}}$ iterations to obtain the adversarial visual inpputs $\mathbf{I}_{\text{adv}}$ which we use to calculate the adversarial loss $\mathcal{L}_{\text{adv}}$. Next, we perform an iterative pixel-space optimization to obtain "negative" samples $\tilde{\mathbf{I}}$. Specifically, this is done using $T_{\text{JEM}}$ iterations using AdamW optimizer. We form our joint energy based loss using the "positive" and "negative" samples. Lastly, we combine these two terms to formulate our overall objective, which we use to update the vision encoder weights. We repeat this process until convergence.

## D   Ablation study

**Architecture**   In table 1, we report the results of both ViT-B-32 and ConvNext base using CLIP-JEM. Notably, despite the two architectures share a similar capacity, the ConvNext FID score is significantly better (by **33.5** points). We hypothesize that this stems from the improved prior that CNN-based architectures serves. Additionally, generated images from ViT contain grid artifact from the patch processing mechanism. Thus, we mainly focus on ConvNext-based models.

**Objectives contribution**   To highlight the importance of combining the adversarial and energy contrastive losses, we train the same model for $1,000$ iterations using (i) contrastive energy loss; (ii) contrastive adversarial loss; and (iii) contrastive energy loss + contrastive adversarial loss. We plot the results of 16 text prompts from MS-COCO for the resulting models in fig. 7. The results indicate that using only the contrastive energy loss does not produce meaningful outputs. In contrast, employing solely the contrastive adversarial loss results in meaningful but unrealistic content. Remarkably, combining the two objectives (CLIP-JEM's approach) leverages the strengths of both CLIPAG and JEMs, yielding superior outcomes.

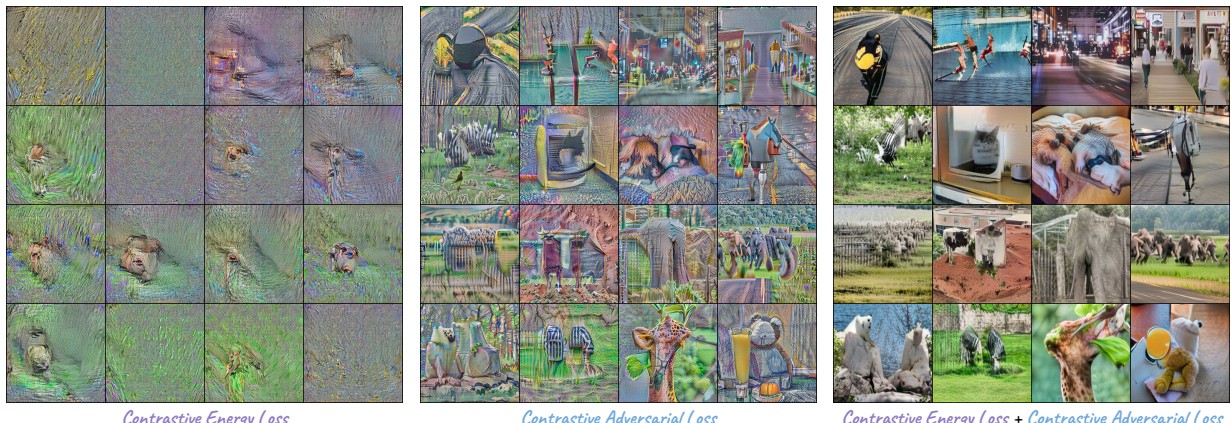

Figure 7: **Objectives ablation study**. CLIP-JEM employs a two objectives: Contrastive Adversarial Loss and a Contrastive Energy Loss. In this figure we ablate the effect of each objective, highlighting the importance of each objective. We plot 16 images based on different textual prompts from MS-COCO after 500 training steps.

## E Connection to Stochastic Gradient Langevin Dynamics

SGLD enables sampling from a distribution $p(\mathbf{x})$ given its score function $\nabla_{\mathbf{x}} \log p(\mathbf{x})$. The stochasticity in this algorithm is important, as applying a simple gradient method would result in sampling the distribution peaks rather than truly sampling the distribution. The iterative update of SGLD in its discrete form is as follows:

$$\boldsymbol{x}_{t+1} = \boldsymbol{x}_t + \frac{\alpha}{2} \nabla_{\boldsymbol{x}_t} \log p(\boldsymbol{x}_t) + \boldsymbol{\epsilon},$$

where $\alpha$ is the step size and $\boldsymbol{\epsilon} \sim \mathcal{N}(\mathbf{0}, \alpha \boldsymbol{I})$ is a Gaussian noise. From eq. (7), the above equation can be also written in terms of energy,

$$\boldsymbol{x}_{t+1} = \boldsymbol{x}_t - \frac{\alpha}{2} \nabla_{\boldsymbol{x}_t} E_\theta(\mathbf{x}_t) + \boldsymbol{\epsilon},$$

and in the joint energy case, the update rule becomes

$$\boldsymbol{x}_{t+1} = \boldsymbol{x}_t - \frac{\alpha}{2} \nabla_{\boldsymbol{x}_t} E_\theta(\mathbf{x}_t, y) + \boldsymbol{\epsilon}.$$

Note that in our algorithm we add a small Gaussian perturbation to the input, prior to the calculation of the gradient (algorithm 1). This introduces a similar, yet different, stochasticity in our approach, compared with SGLD. To better understand this, we expand the following Taylor series:

$$\nabla_{\mathbf{x}} E_\theta(\mathbf{x} + \epsilon, y) \approx \nabla_{\mathbf{x}} E_\theta(\mathbf{x}, y) + \nabla_{\mathbf{x}}^2 E_\theta(\mathbf{x}, y)^T \boldsymbol{\epsilon}$$
$$+ \mathcal{O}(\|\boldsymbol{\epsilon}\|_2^2).$$

As our noise is of small variance, the higher order term can be neglected,

$$\nabla_{\mathbf{x}} E_\theta(\mathbf{x} + \epsilon, y) \approx \nabla_{\mathbf{x}} E_\theta(\mathbf{x}, y) + \nabla_{\mathbf{x}}^2 E_\theta(\mathbf{x}, y)^T \boldsymbol{\epsilon},$$

making it similar to the SGLD noisy update step, as the term $\nabla_{\mathbf{x}}^2 E_\theta(\mathbf{x}, y)^T \boldsymbol{\epsilon}$ is a linear transformation of the Gaussian noise, resulting in a colored version of a Gaussian distribution, governed by the Hessian of the energy function.

We demonstrate the randomness introduced by this mechanism using our sampling process in fig. 8. Unlike to SGLD, we utilize an AdamW optimizer for the update step for the generated sample. Thus, we do not use the raw gradient as in SGLD since our optimizer has an adaptive learning rate mechanism. This results in a different effective step size per iteration ($\alpha$ in SGLD). AdamW also enables momentum term, which

A cozy cabin in the woods with smoke coming from the chimney

A medieval knight standing in front of a castle

A group of astronauts exploring a distant planet

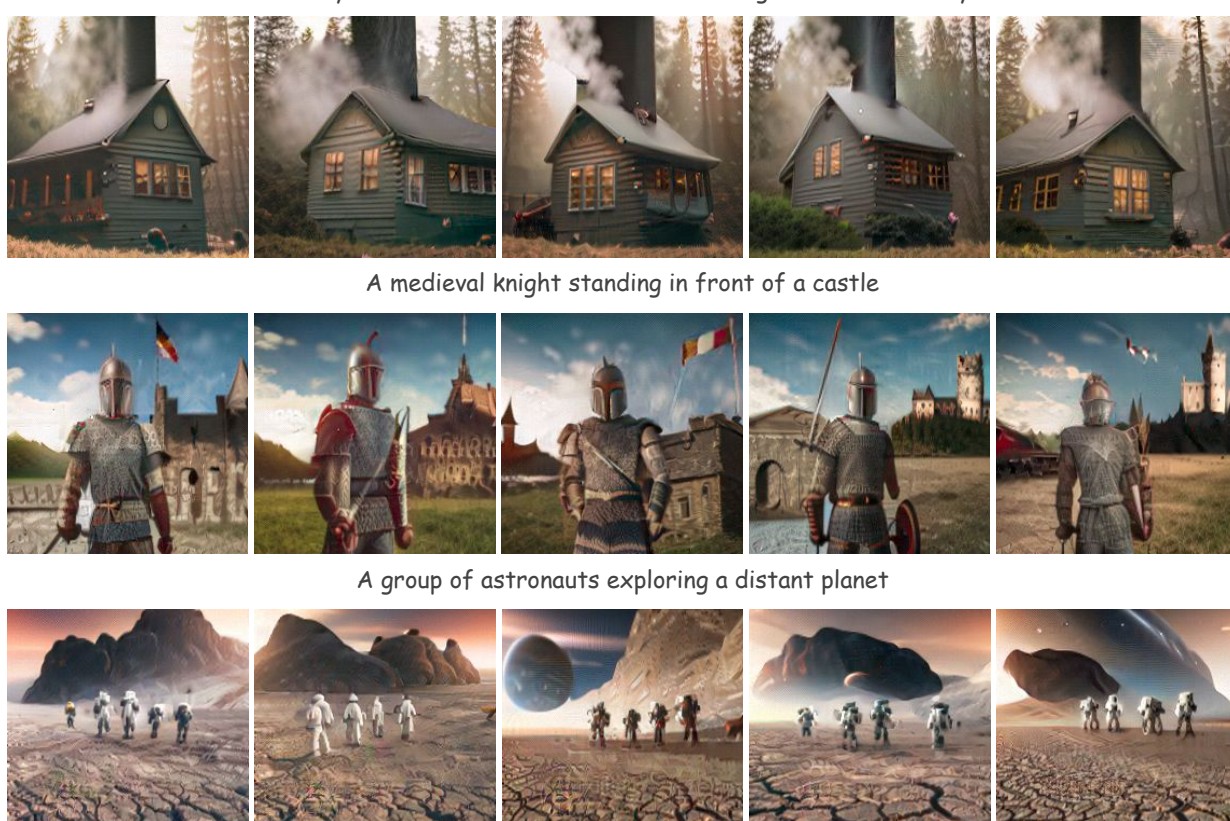

Figure 8: **Stochasticity demonstration of CLIP-JEM sampling**.

introduces bias to the gradients. During CLIP-JEM training, we utilize the momentum as it significantly stabilizes the training. However, at inference time, we do not employ momentum, resulting in an unbiased version of the gradients.

## F    Additional results

We provide additional qualitative results for both improving CLIP-based generative frameworks and text-to-image generation. For the former, we provide the results using the same prompts with augmentations (figs. 10 and 11). As can be seen, seamlessly replacing CLIP with CLIP-JEM leads to improved results with and without augmentations. Notably, CLIP-JEM results without augmentations are comparable to the ones of CLIP with augmentation, offering a substantial reduce of computational overhead. For the latter, we provide more generated image (fig. 9) and demonstrate the stochasticity of CLIP-JEM(fig. 8).

## G    Positioning CLIP-JEM  within the Landscape of Generative Models

**Comparison with Diffusion Models**    While CLIP-JEM  does not achieve state-of-the-art FID scores in text-to-image generation (see table 1), it offers several notable advantages. It exhibits superior compositional generalization, outperforming Stable Diffusion v1.4 and v2 on the T2I-CompBench benchmark (Gadre et al., 2024), as well as surpassing compositionality-oriented diffusion-based methods. This performance stems from its joint energy–contrastive adversarial training, which enhances the alignment between textual prompts and generated images. Unlike diffusion pipelines that rely on multi-step denoising in pixel or latent spaces

- necessitating a learned decoder and intricate noise scheduling - CLIP-JEM samples directly in pixel space through a straightforward iterative gradient-following process. This approach yields energy values proportional to sample likelihoods, facilitating explicit probabilistic ranking and selection of outputs. We leverage this property to develop a robust text-to-image alignment metric (see section 4.3) and to convert unconditional diffusion models into text-conditioned generators. Furthermore, CLIP-JEM 's energy-based formulation allows for seamless "plug-and-play" integration into CLIP-based synthesis workflows; for example, replacing the CLIP encoder in CLIPDraw with CLIP-JEM results in more semantically faithful renderings under the same optimization loop (Frans et al., 2022).

**Comparison with Architectures Employing CLIP Image Encoders**  Leading text-to-image systems incorporate CLIP's image encoder alongside distinct generative backbones rather than utilizing it to generate pixels directly. DALL · E 2 (Ramesh et al., 2022) employs a diffusion prior to map CLIP text embeddings to CLIP image embeddings, followed by a modified GLIDE decoder for pixel synthesis. DiffusionCLIP (Kim et al., 2022) integrates a CLIP-based contrastive loss into the reverse diffusion trajectory of pretrained diffusion models to enable zero-shot, text-guided image manipulation. VQGAN-CLIP (Crowson et al., 2022) pairs a pretrained VQGAN generator with CLIP's image encoder to score and steer pixel-space updates via gradient descent. Recently, BLIP3-o (Chen et al., 2025) utilizes the CLIP image encoder in conjunction with a large language model and a diffusion transformer to generate images. In contrast, CLIP-JEM constructs a generative model directly from CLIP itself, eliminating the need for an additional generative model.

## H   Limitation and Futrue Work

Our current work exhibits two main limitations. First, CLIP-JEM shows limited diversity in its generated samples. The primary source of stochasticity comes from the random initialization process, and while we experimented with noise injection during optimization steps, we maintained a small noise coefficient for stability. This resulted in generated images sharing similar visual characteristics for the same prompt, as evident in fig. 8. Future work could explore the enhancement of sample diversity through increased noise levels during optimization or alternative initialization strategies. Second, CLIP-JEM underperforms on tasks involving spatial relationships, as demonstrated in table 2. This limitation stems from CLIP's inherent constraints in spatial compositionality, as our model initializes from CLIP weights. This aligns with the well-documented limitations of CLIP in handling spatial relationships. Future research could focus on addressing these spatial understanding capabilities, potentially through architectural modifications or specialized training objectives that better capture spatial information.

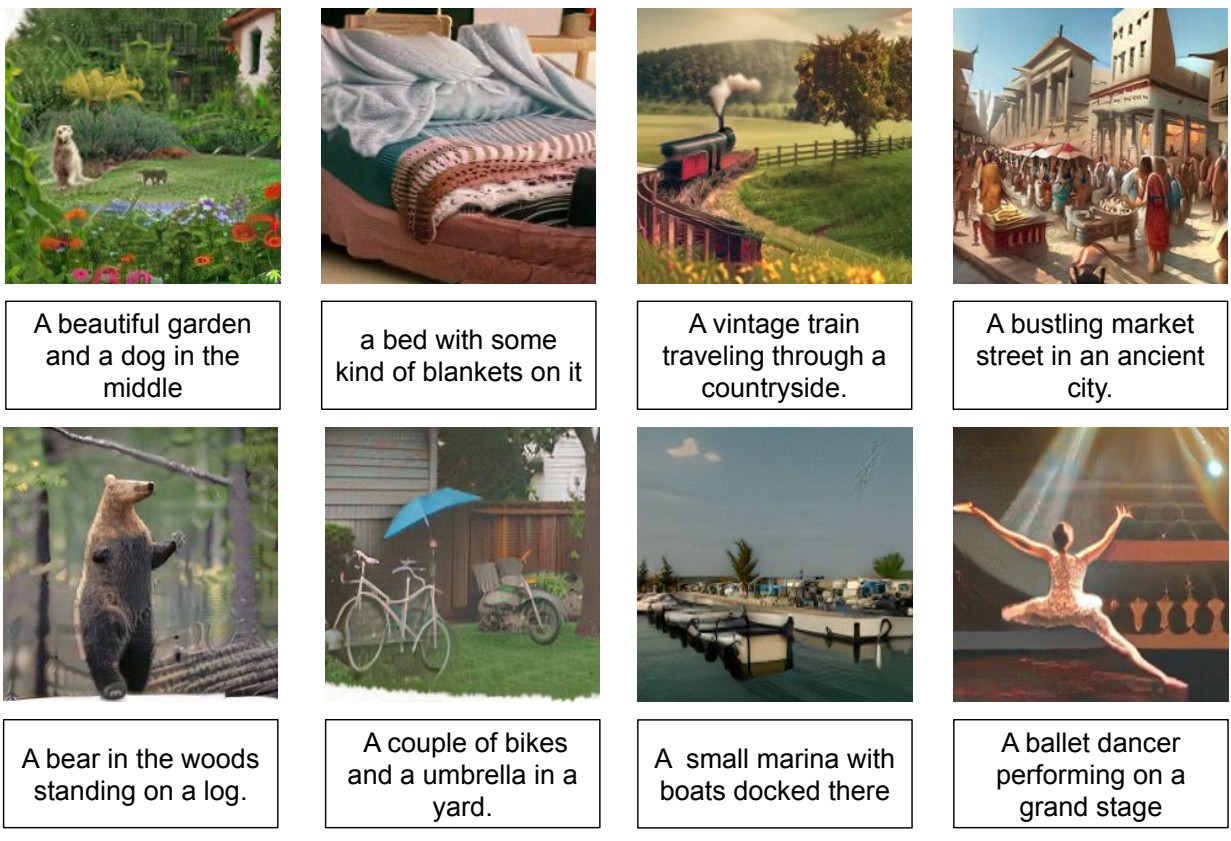

Figure 9: **Additional qualitative results**. Samples generated by CLIP-JEM using ConvNext XXL.

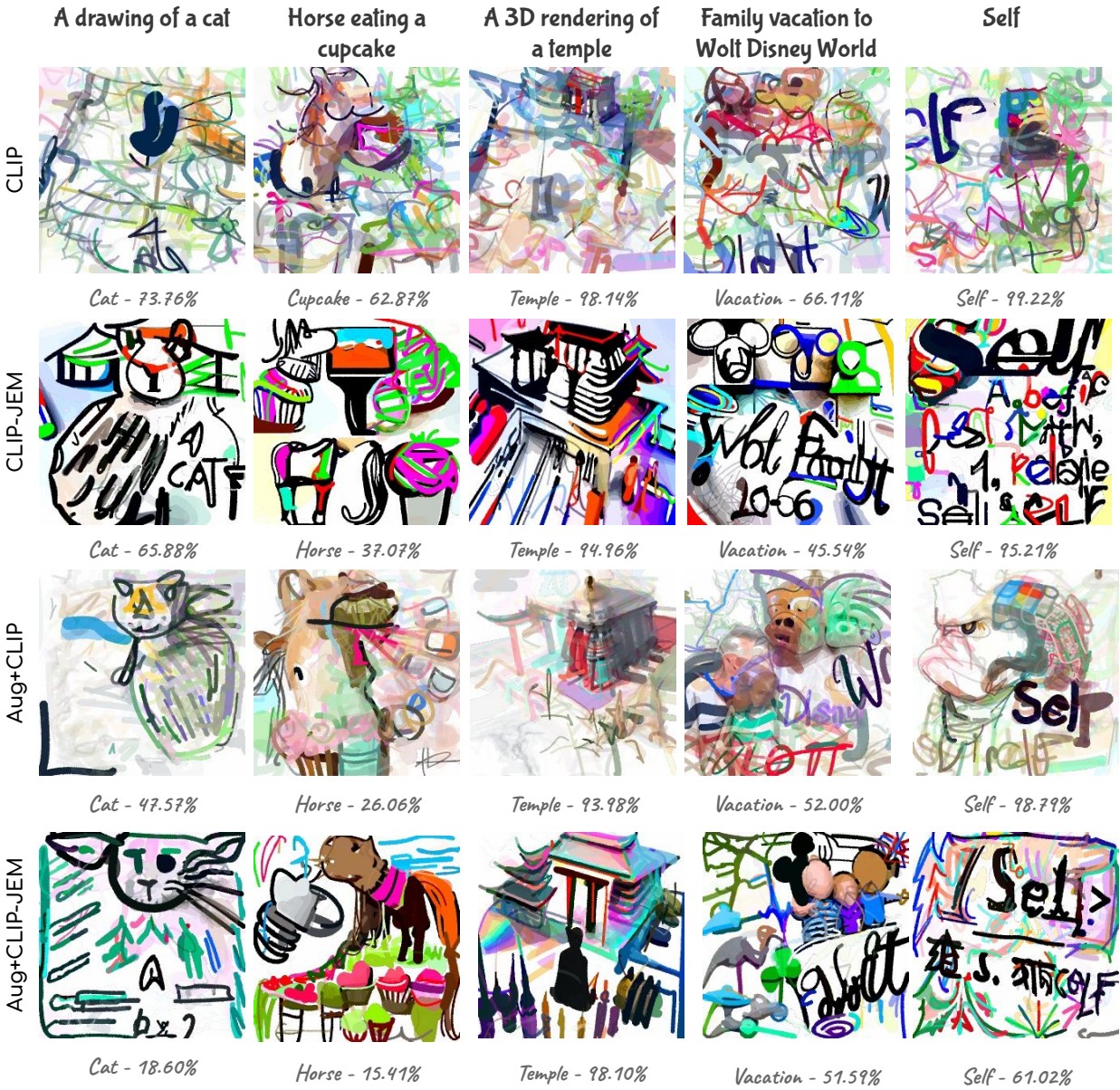

Figure 10: **CLIPDraw results**. We provide the results of CLIP and CLIP-JEM with and without augmentations.

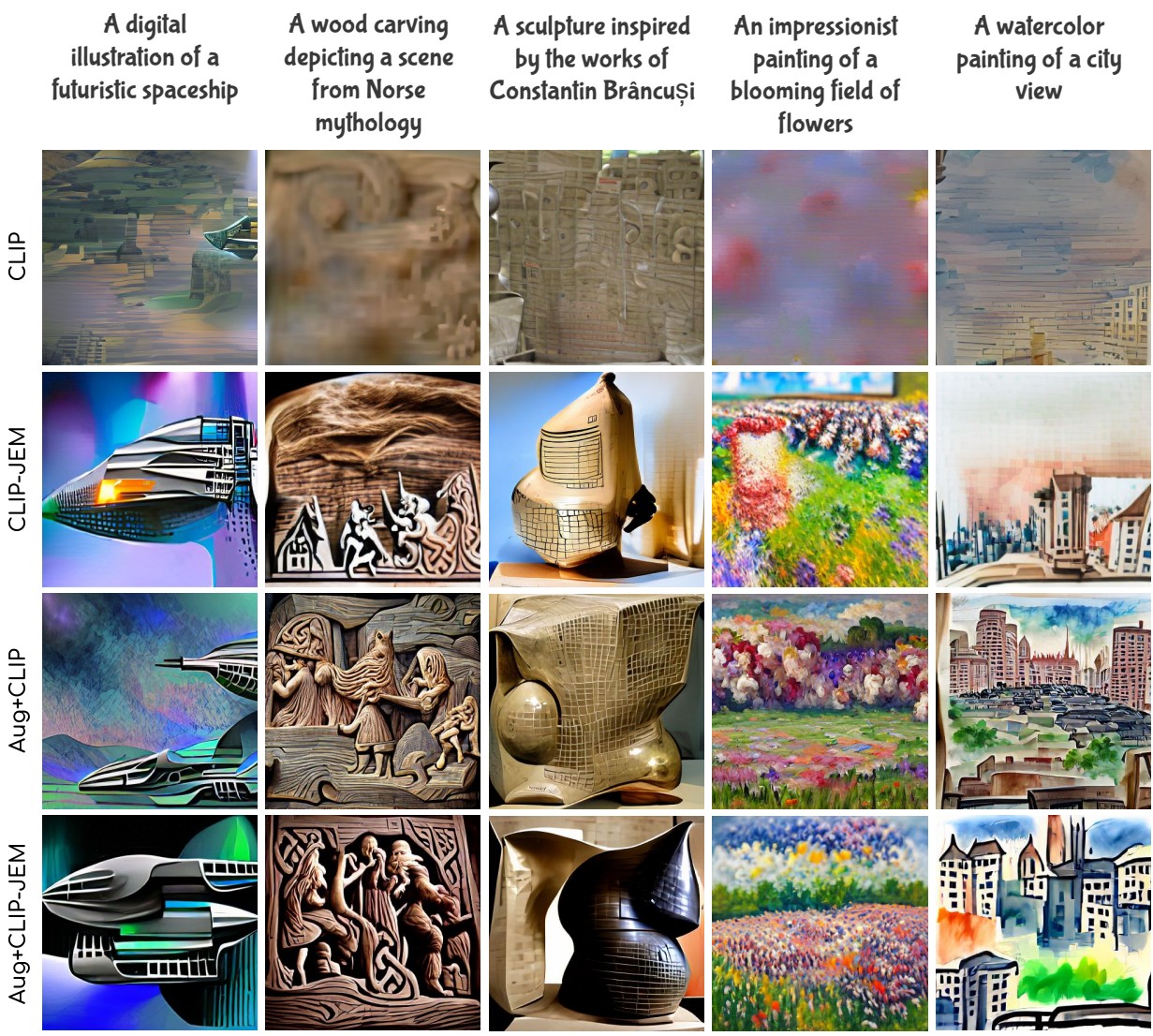

Figure 11: **VQGAN+CLIP results**. We provide the results of CLIP and CLIP-JEM with and without augmentations.

