# OpenReview forum: "Text-to-Image Generation Via Energy-Based CLIP"
_TMLR — Accepted by TMLR_

### Review · Reviewer_3D78 · 2025-04-12

**Summary Of Contributions:**

The paper proposes CLIP-JEM to extend Joint-Energy-Models (JEMs) to visual-language domain using CLIP. Specifically, this is achieved by a combination of one generative and one discriminative objective. The generative objective extends the JEMs to a slightly different visual-language version; the discriminative one adapts adversarial training to the contrastive loss similar to the previous CLIPAG. The authors perform text-to-image generation experiments on MS-COCO, and test compositionality on T2I CompBench. They also used CLIP-JEM to improve the existing  CLIP-based generative frameworks, and propose to use CLIP-JEM as an evaluation metric with better sensitivity to perceptual quality with added noises.

**Audience:**

Yes

**Broader Impact Concerns:**

A Broader Impact Statement would probably be helpful despite the methodology focus, as this is a work on the generative models, and may produce unlicensed and potentially harmful content.

**Claims And Evidence:**

Yes

**Requested Changes:**

In addition to the W1, W2, and W3 above, the reviewer is also curious about one specific setting of ablation experiments. Specifically, the authors mention that CILPAG is heavily reliant on multiview augmentation, claiming that the proposed CLIP-JEM does not, and show a comparison with the case w/o multiview augmentation. However, does further adding multiview augmentation will help CLIP-JEM? Why or why not?

**Strengths And Weaknesses:**

S1: The paper is generally well-structured and easy to follow.

S2: The methodology design is well-motivated and shows valid improvement compared to related baselines such as CLIPAG.

S3: Experiments somewhat demonstrate and emphasize the potential versatile usage of the proposed CLIP-JEM, such as the compositionality and the usage as an evaluation metric.

W1: The existing evaluations lack the discussion on the diversity of generated samples. While the authors emphasize the general quality and alignment w.r.t. the text prompt, the ability to generate diverse images based on a single prompt seems to be excluded from current discussions. From the existing qualitative samples from Fig.8, the generated images given a specific prompt appear to be visually similar.

W2: The exact formats of the contrastive adversarial objective are missing from Sec. 3.2. It is recommended to explicitly write down the equations as in Sec. 3.1 to improve the clarity.

W3: The paper lacks specific details on the computational cost for training, while Tab. 4 and Tab. 5 show some training implementation details and sampling cost, what is the actual training time cost for those variants?

---

> ### Author Response · Authors · 2025-06-06
> **Authors reply**
>
> We thank the reviewer for the thoughtful and constructive feedback. Please find our detailed responses below:
>
> - **Diversity of Generated Images**: We agree that the diversity illustrated in Figure 8 is somewhat limited. In CLIP-JEM, the primary source of stochasticity comes from the random initialization. While we also explored injecting noise at each optimization step, we employed a small coefficient, limiting the impact of this source of randomness (please refer to Algorithm 1). Greater diversity could be achieved by increasing the noise level or adopting alternative initialization strategies. However, as the primary focus of this work is to demonstrate the capability of generating realistic images from text using an adversarial joint energy model, we leave such enhancements in diversity for future work.
>
> - **Contrastive Adversarial Objective**: We have added a detailed explanation of the contrastive adversarial objective in Section 3.2 of the revised manuscript.
>
> - **Computational Cost**: Additional details regarding computational cost have been included in the appendix.
>
> **Multiview Augmentation**: In our experiments, multiview augmentation did not yield measurable improvements, either qualitatively or quantitatively. We hypothesize that its main benefit in CLIPAG stems from guiding generation steps that stray far from the image manifold, as CLIPAG is trained on adversarial samples initialized near real images (with a constrained adversarial budget). In contrast, CLIP-JEM is trained to generate realistic images from random initializations and thus produces meaningful gradients even when samples are far from the manifold. We believe this difference explains why multiview augmentation does not offer similar benefits in our setting.

---

### Review · Reviewer_2vtY · 2025-05-19

**Summary Of Contributions:**

The paper introduces CLIP-JEM, a novel joint energy-based model extending to text-to-image generation by combining both generative and discriminative training objectives. It defines an energy function using the cosine similarity within CLIP's embedding space, which assigns low energy to real image-text pairs and high energy to mismatched pairs. CLIP-JEM significantly outperforms prior methods like CLIPAG in generating realistic images without relying on extensive augmentations, demonstrating superior performance on challenging compositional benchmarks.

**Audience:**

Yes

**Broader Impact Concerns:**

Since this paper is proposing a generative model, it would be useful to discuss how to prevent misuse of the proposed method.

**Claims And Evidence:**

Yes

**Requested Changes:**

See previous section point 2 and 3.

**Strengths And Weaknesses:**

### Strengths
1) The idea of revesting JEM to build a modern generative model is interesting.
2) The idea of combining adversarial training and generative objective is novel.
3) Empirical results are promising.

### Weaknesses
1) Section 2.1 is not clear:
- "The model converges when it cannot distinguish between positive and negative samples.": The model assigns energy to data points. How does it distinguish between positive and negative samples?
2) It would be interesting to discuss the advantage of energy model over diffusion model (or flow-based generative models), which has a similar sampling function and has strong performance.
3) Some previous/concurrent works also leverage CLIP feature to build a generative model, e.g., DALLE-2 or BLIP3-o. It would be helpful to discuss the relationship of this work w.r.t. them.

---

> ### Author Response · Authors · 2025-06-06
> **Authors reply**
>
> We thank the reviewer for the constructive feedback and we have revised the paper accordingly.
> Specifically, we added a clarification in section 2.1 and the requested discussions in appendix G.

---

### Review · Reviewer_9mgW · 2025-05-20

**Summary Of Contributions:**

The paper introduces CLIP-JEM, a novel approach that extends Joint Energy Models (JEMs) to the multimodal vision-language domain using CLIP. The key contributions are:

1. CLIP-JEM is a novel approach extending Joint Energy Models to the vision-language domain using CLIP.

2. Improving Text-to-Image Generation via pixel-space optimization

3. Enhancing text-based guidance capabilities and converting unconditional diffusion models into text-guided ones with just 25 DDIM steps

4. CLIP-JEM can serve as an improved CLIP-Score evaluation metric for text-based image editing.

The method achieves competitive results on MS-COCO and T2I-CompBench, demonstrating strong compositional understanding despite its smaller size compared to models like SD v2.

**Audience:**

Yes

**Broader Impact Concerns:**

No Broader Impact Concerns

**Claims And Evidence:**

Yes

**Requested Changes:**

1. Please provide some explanations about why the model performs unsatisfying in the spatial tasks.

2. Some quantitative evaluation results of the guidance are expected.

**Strengths And Weaknesses:**

Strengths:
1. Creatively combines the strengths of EBMs (generative flexibility) and CLIP (multimodal alignment).

2. CLIP-JEM achieves impressive results across multiple evaluation benchmarks.

3. The authors have validated that the model's versatility across multiple applications: direct text-to-image generation, guidance for other models, and as an evaluation metric.

Weaknesses:

1. The model seems to struggle with spatial relationships compared to other methods. Is there any explanation of this situation?

2. Quantitative evaluation of guidance: The guidance capabilities are primarily demonstrated through qualitative examples, but more quantitative evaluation would strengthen this claim.

---

> ### Author Response · Authors · 2025-06-06
> **Authors reply**
>
> We thank the reviewer for the constructive feedback.
>
> - **Performance in spatial relationships**: We agree that CLIP-JEM underperforms on spatial relationships, as shown in Table 2. We attribute this to the well-documented limitations of CLIP in spatial compositionality, given that CLIP-JEM is initialized from CLIP weights. We cite relevant work on this limitation in the Compositionality paragraph of Section 4.1.
>
> - **Guidance capabilities**:  Our intention was to demonstrate the conceptual feasibility of transforming unconditional diffusion models into text-conditional ones using CLIP-JEM guidance. While we agree that a quantitative evaluation could provide additional insights, we believe it is not essential in this case. The original diffusion samples are unconditional and therefore naturally exhibit poor text-to-image alignment. In contrast, the CLIP-JEM-guided results show a clear qualitative improvement, introducing guidance capabilities that were entirely absent before. As such, we consider the qualitative comparison (Figure 4) sufficient to support the validity of our approach.

---

### Decision · Action_Editor_QQy3 · 2025-07-10

**Recommendation:** Accept with minor revision

**Additional Comments:**

While most concerns have been addressed in the rebuttal, I do agree with the reviewers that the experimental evaluation can be improved, e.g., it is beneficial to conduct more experiments on large-scale settings, and provide enough discussions on the limitation of the proposed method, e.g., the sample diversity issues should be more well addressed.

**Audience:**

Yes

**Audience Explanation:**

Researchers in generative models could find the work interesting as it is different from the SoTA methods.

**Claims And Evidence:**

Yes

**Claims Explanation:**

This paper incorporates energy-based model and CLIP to construct a multi-modal generative model. Through joint generative and discriminative training, the proposed method is shown to be able to generate realistic images and achieve competitive results on some compositionality benchmarks, with very few learnable parameters.

The reviewers initially raise some concerns such as the presentation, the lack of a clear demonstration of the advantages of the proposed method, the ability to deal with spatial relationships, limited quantitative evaluations, especially on the demonstration of the diversity of generated images. The authors provide revised discussion and additional clarifications in the rebuttal, which the reviewers are generally satisfied. All reviewers believe the submission is supported by enough evidence for acceptance.